# Demystifying Prompts in Language Models via Perplexity Estimation

**Hila Gonen**[1,2]    **Srini Iyer**[2]    **Terra Blevins**[1]    **Noah A. Smith**[1,3]    **Luke Zettlemoyer**[1,2]

[1]Paul G. Allen School of Computer Science & Engineering, University of Washington
[2]Meta AI Research    [3]Allen Institute for Artificial Intelligence
hilagnn@gmail.com
sviyer@meta.com
{blvns,nasmith,lsz}@cs.washington.edu

## Abstract

Language models can be prompted to perform a wide variety of tasks with zero- and few-shot in-context learning. However, performance varies significantly with the choice of prompt, and we do not yet understand why this happens. In this paper, we analyze the factors that contribute to this variance and establish a new empirical hypothesis: the performance of a prompt is predicted by the extent to which the model is familiar with the language it contains. Over a wide range of tasks, we show that the lower the perplexity of the prompt, the better it is able to perform the task, when considering reasonable prompts that are related to it. As part of our analysis, we also devise a method to automatically extend a small seed set of manually written prompts by paraphrasing with GPT3 and backtranslation. This larger set allows us to verify that perplexity is a strong predictor of the success of a prompt and we show that the lowest perplexity prompts are consistently effective.

## 1 Introduction

Language models can be prompted to perform a wide range of zero- and few-shot learning tasks (Brown et al., 2020; Schick and Schütze, 2020). However, there is significant variance in the performance of seemingly similar prompts (Chen et al., 2022): for AG News (Zhang et al., 2015), we find an over 30 point accuracy gap between different manually curated prompts (see Table 1) on OPT 175B (Zhang et al., 2022). Despite efforts to improve prompt engineering (Shin et al., 2020; Li and Liang, 2021; Gao et al., 2021), it is still challenging to develop high-quality prompts for new tasks, and little is known about why this phenomenon occurs.

We are interested in understanding what makes some prompts better than others, and using this understanding to create better prompts for given tasks and models. We hypothesize that the lower the perplexity of a prompt is, the better its performance

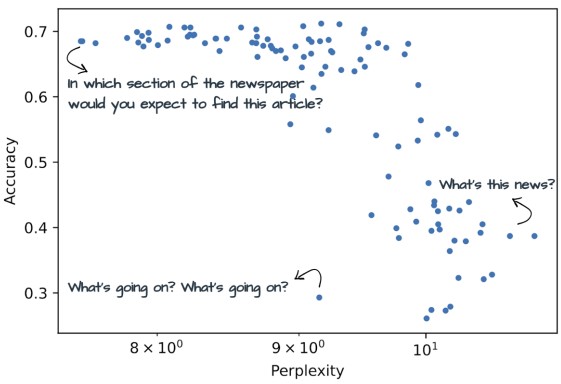

Figure 1: Accuracy vs. perplexity for the AG News dataset with OPT 175B. The $x$ axis is in log scale. Each point stands for a different prompt.

on the task will be, when considering reasonable prompts that are related to the task. This is based on the intuition that the more frequently the prompt (or very similar phrases) appears in the training data, the more the model is familiar with it and is able to perform the described task. We refrain from using the training data directly as it is often unavailable, expensive to search due to its size, and hard to use for approximate matching of similar prompts. Instead, we focus on the perplexity of the prompt as a proxy for its occurrences in the data.

To enable more complete analysis, we automatically expand the set of manually created prompts for the task by paraphrasing, resulting in a much larger and diverse set of prompts. We focus on prompts in English that reasonably describe the task for two reasons: (a) our main motivation is to understand what lies under the variance of performance in this type of prompt; (b) we aim to devise a useful method for creating prompts that are consistently effective, that could be easily adopted and interpreted by future, potentially non-expert users.

We show empirically that our hypothesis holds across a diverse set of tasks (including classification and word prediction), models, and model

sizes, providing us some insights about the underlying mechanism of prompting (see Figure 1). As a result, we devise a method, SPELL (Selecting Prompts by Estimating LM Likelihood), for creating prompts in an informed manner. We show that using SPELL to choose prompts results in less variability in performance as well as in accuracy gains (1.8 accuracy points with OPT and 2.3 accuracy points with Bloom on average). Importantly, our method does not require labels at all, only a small sample of inputs for the task.

Our contributions can be summarized as follows: (a) we formalize the notion that better familiarity of the model with the prompt correlates with better performance (Section 2); (b) we automatically elaborate a given set of seed prompts using paraphrasing (Section 3); (c) we establish experimentally the hypothesis that lower perplexity of the prompt correlates well with better performance (Section 5); (d) we devise a method to create a more consistent set of prompts, that also improve results even with no labels for the task (Section 7).

## 2   Why are prompts not all created equal?

Despite the popularity of prompting as a method for using language models (Shin et al., 2020; Li and Liang, 2021; Gao et al., 2021), the cause for the different behavior of various prompts remains unclear so far. Table 1 shows four example prompts for a news topic classification task (AG News) and their respective accuracies when used to prompt OPT 175B (Zhang et al., 2022). The accuracy gap between the different prompts is not trivial, and it is not possible to predict from the prompts alone.

| Prompt | Accuracy |
|---|---|
| What is this piece of news regarding? | 40.9 |
| What is this article about? | 52.4 |
| What is the best way to describe this article? | 68.2 |
| What is the most accurate label for this news article? | 71.2 |

Table 1: Example prompts for the task AG News (news classification) that vary considerably in accuracy.

We propose that the more frequently a prompt appears in some variation in the data, the better it works for the task. The intuition behind this is that a sequence that is more expected by the model is more likely to aid the model to extract the relevant information. However, this premise is hard to measure accurately: most language models use huge amounts of training data (e.g., OPT uses a corpus of roughly 180B tokens, and Bloom uses

roughly 366B tokens), and in addition, this training data is not always publicly available (e.g., GPT3; Brown et al. 2020). Our initial attempts to estimate exact-match occurrences of prompts in the data resulted in very sparse counts, which led us to look for a softer formalization.[1]

Instead of considering the training data directly, we propose to focus on the *perplexity* of the prompt as a proxy for its occurrences in some form in the data – essentially indicating to what extent the model expects this prompt. This perplexity-based framing helps to avoid the challenge of exact match in the data, and takes into account variations of the prompt that the model is also exposed to and might be influenced by. In addition, it helps overcome the challenges mentioned above as it requires neither access to the pretraining data (which is not always publicly available for LMs) nor matching over huge amounts of text.

**Hypothesis: lower perplexity correlates with better performance.**   We hypothesize that on average, lower-perplexity prompts perform better. We are interested in establishing this hypothesis by experimentally showing a significant negative correlation between the perplexity of the prompt and its performance on the task, across a diverse set of tasks and models.

We define the perplexity of the prompt as the perplexity of the full prompt sequence, including the input itself, and without the label, averaged over 1,000 examples (see Section 4 for details). The input is a part of the prompt in the case of the word prediction tasks by design (e.g., "The opposite of the word **good** is"). Inclusion of the task input as part of the prompt for classification tasks as well is intentional: we want to ground the prompt to the task (without the input, we are testing the hypothesis that lower perplexity prompts across all tasks work better on every task). The label is not considered a part of the prompt and is not taken into consideration when computing the prompt. In practice, this also results in a huge advantage of our method, SPELL (Section 7), which aims to find better prompts—it does not require any labels.

For performance measures, we use the log-likelihood score assigned by the model to the correct label given that prompt. We choose this metric

---

[1] We experimented with the task of AG News (see Section 4.1), and looked for all of its prompts (using exact match) in the OPT training data. Indeed, only 9/108 of the prompts appear in the training data. Such sparse counts do not allow for any useful or reliable analysis of prompt behaviour.

over accuracy as it gives a more fine-grained distinction between prompts and because accuracy can be unstable, as explained in more detail in Section 4. For classification tasks, we also report correlation with accuracy, which is the main evaluation metric for this type of task.

## 3 Automatic Expansion of Seed Prompts

We are interested in expanding our pool of prompts in order to: (a) have a more diverse set of prompts, making it more likely to find a better prompt for our task, and (b) support better analysis to validate our prompt quality hypothesis. In this section, we describe our method for automatically expanding a seed set of manually created prompts using paraphrasing.

**Step 0: Creating a seed set of manually-written prompts**  We first write/collect a small set of human written prompts that describe the task. For classification tasks we assume that the input appears before the prompt, with no choices appearing as part of the prompt (to help in smooth paraphrasing of the prompt itself).

**Step 1: Paraphrasing with GPT3**  We use the text-davinci-002 version of GPT3 (Brown et al., 2020) to generate paraphrases for each of the manual prompts in our seed set. We prompt it with a *meta-prompt* for paraphrasing to generate variations of one of our seed prompts. An example of such a meta-prompt is: *Write a paraphrase for the following sentence:* <seed prompt> *Paraphrase:*. The 7 meta-prompts used in this step are listed in Table 2.

We choose GPT3 as our paraphrasing model because of its well-documented generation abilities. This is also to ensure that there is a separation between the model we use to create the prompts and the models we use to rank them (OPT and Bloom, see Section 4 for details), to avoid confounding the experimental setup.

**Step 2: Paraphrasing using backtranslation**  Our second step takes as input the paraphrases from GPT3 (in addition to the seed set of prompts) and translates them into different languages and back into English to get additional prompt paraphrases (Wieting et al., 2017). We use a set of 8 languages available in the NLLB translation model (Costa-jussà et al., 2022) that are relatively high resource

and close to English,[2] to reduce the risk of noise. Since we aim to get about 100 prompts per task, we add 8 additional languages[3] in the case where the basic 8 languages yielded too few alternatives. For word prediction tasks, we use the sequence of the created prompt up to the index of the label, not including the label, for example: *The word "dog" in French is "*. Depending on the task, we enforce the existence of specific words (e.g., the name of the language, and the source word, in word-level translation) or enforce the prompt to be a question.

**Examples and Statistics**  Table 4 lists all 4 manually created prompts we use for the AG News task (news classification), alongside a few sampled prompts created automatically using our method. As was typically the case, we are able to get prompts that are rather different in phrasing and structure from those included in the seed set.

The statistics of the prompts in the manually created seed set (Step 0) as well as the prompts after Step 1 and Step 2 for each task (see Section 4.1 for details about the tasks) are detailed in Table 3.

## 4 Experimental Setup

### 4.1 Models, Tasks and Datasets

We study four auto-regressive models: OPT (Zhang et al., 2022) of different sizes (1.3B, 30B, 175B parameters), all trained mainly on English,[4] and Bloom (176B parameters; Luccioni et al. 2022), which is trained on 46 natural languages and 13 programming languages. We experiment with two types of tasks: word prediction tasks and classification tasks, as detailed below.

**Word Prediction Tasks**  The first task in this category is **word-level translation**. Given a source word in English and a target language, we expect the model to predict the correct translation. For this task we use NorthEuraLex[5] (Dellert et al., 2019), a lexical database providing translations of 1016 words into 107 languages. We experiment with 9 languages that use the Latin script. For Bloom, we use 5 additional languages that do not use the

---

[2]Danish, German, Italian, French, Dutch, Portuguese, Swedish, Spanish.

[3]Norwegian, Romanian, Catalan, Turkish, Ukrainian, Polish, Russian, Arabic.

[4]As stated in the paper, the training corpora were previously collected or filtered to contain predominantly English text, but a small amount of non-English data is still present within the corpus via CommonCrawl.

[5]http://northeuralex.org/

| Meta prompts |
|---|
| Write a paraphrase for the following sentence: <seed-prompt> Paraphrase: |
| <seed-prompt> Paraphrase: |
| Write a likely paraphrase of the text: <seed-prompt> Paraphrase: |
| Write a sentence similar to the following one: <seed-prompt> Paraphrase: |
| Paraphrase the following sentence: <seed-prompt> Paraphrase: |
| Write a variation of this sentence: <seed-prompt> |
| How would you say the following sentence in a different way? <seed-prompt> |

Table 2: Meta prompts used in Step 1 of our method for paraphrasing using GPT3.

| Task | # Step 0 | # Step 1 | # Step 2 |
|---|---|---|---|
| Word-Level Translation | 12 | 59 | 118 |
| Antonyms | 12 | 85 | 176 |
| GLUE Cola | 4 | 27 | 144 |
| Newspop | 13 | 43 | 119 |
| AG News | 4 | 23 | 108 |
| IMDB | 10 | 45 | 178 |
| DBpedia | 8 | 23 | 103 |
| Emotion | 4 | 14 | 94 |
| Tweet Offensive | 5 | 41 | 119 |

Table 3: Number of prompts for the different tasks: prompts after step 0 (creating prompts manually), prompts after step 1 (GPT3 paraphrasing), and prompts after step 2 (backtranslation).

Latin script (since Bloom is multilingual). Note that only 5 of the languages we experiment with are officially covered by Bloom.[6]

We also consider **antonym prediction** where, given a word, the model is expected to predict its antonym. For this task, we use data from Kaggle,[7] which is based on WordNet (Miller, 1995). We choose 1,000 word pairs at random.

**Classification Tasks** We choose classification tasks from Huggingface Datasets,[8] with an attempt to have a set of diverse tasks that use relatively short inputs, with some prompts available in Prompt-Source (Bach et al., 2022):[9] (a) **GLUE Cola** (grammaticality; Warstadt et al. 2018); (b) **Newspop** (news classification; Moniz and Torgo 2018); (c) **AG News** (news classification; Zhang et al. 2015); (d) **IMDB** (movie review classification; Maas et al. 2011); (e) **DBpedia** (topic classification; Lehmann et al. 2015); (f) **Emotion** (classification to emo-

tions; Saravia et al. 2018); (g) **Tweet Offensive** (classification to offensive vs. not offensive tweets; Barbieri et al. 2020). We use 1,000 random examples from each dataset.

The full set of manual prompts is listed in Section A in the Appendix. In these tasks, the prompt follows the input, and at the end of each prompt we add the choices of classes (i.e., we provide the possible labels explicitly in the prompt by listing the possible answers as defined by the dataset itself.): *"Choices: X, Y, Z. Answer:"* as we find it helps in terms of accuracy. Defining the label space likely helps in our zero-shot setting because there are no previous demonstrations from which the model can learn the possible classes. Additionally, adding class options to the prompt helps to reduce the effect of the surface form competition (Holtzman et al., 2021). The option of generating the answer and comparing it with the gold label was not reasonable here, since we cannot expect the model to generate the exact label as the first choice often enough.

### 4.2 Implementation Details

In all experiments we evaluate zero-shot performance. To avoid noise when computing perplexity, we instantiate the prompts with 1,000 examples of the dataset, compute the perplexity of the prompt with each example, and calculate the average across all instantiated prompts.

To estimate the performance of the prompt, we look at two measures: (a) the language model score (log probability) of the correct label, averaged across 1,000 examples; (b) the accuracy on the task, computed over the 1,000 examples. To compute accuracy, for each example we score all classes and choose the highest ranking class as the prediction of the model. The score of a label of multiple tokens is defined by the sum of the token

---

[6]Basque, French, Portuguese, Spanish, and Arabic.
[7]https://www.kaggle.com/datasets/duketemon/antonyms-wordnet
[8]https://huggingface.co/docs/datasets/index
[9]https://github.com/bigscience-workshop/promptsource

| All Manually Created Prompts | Examples of Similar Automatically Created Prompts |
|---|---|
| What label best describes this news article? | What's the most accurate label for this news article? |
| What is this piece of news regarding? | What does this piece of news concern? |
| Which newspaper section would this article likely appear in? | In what section of the newspaper could this article be published? |
| What topic is this news article about? | What category does this article fall into? |

Table 4: Prompts for the task AG News (news classification): the manually created prompts and a sample of automatically created prompts using our method.

| Model | Task | Perplexity-score corr. | | Perplexity-acc corr. | | Avg Acc | Acc 50% |
|---|---|---|---|---|---|---|---|
| | | **Pearson** | **Spearman** | **Pearson** | **Spearman** | | |
| OPT 175B | Antonyms | **-0.41 | **-0.53 | – | – | – | – |
| | GLUE Cola | -0.15 | -0.14 | -0.04 | -0.02 | 47.7 | 57.1 |
| | Newspop | *-0.24 | **-0.26 | *-0.20 | -0.18 | 66.4 | 72.9 |
| | AG News | **-0.63 | **-0.68 | **-0.77 | **-0.81 | 57.5 | 68.7 |
| | IMDB | **0.35 | **0.40 | 0.14 | *0.20 | 86.2 | 91.0 |
| | DBpedia | **-0.50 | **-0.44 | **-0.51 | **-0.42 | 46.7 | 55.2 |
| | Emotion | -0.14 | -0.19 | **-0.30 | **-0.32 | 16.4 | 23.0 |
| | Tweet Offensive | *-0.19 | 0.07 | 0.18 | *0.23 | 51.3 | 55.8 |
| Bloom 176B | Antonyms | **-0.37 | **-0.23 | – | – | – | – |
| | GLUE Cola | 0.07 | 0.11 | **-0.25 | **-0.26 | 55.5 | 65.6 |
| | Newspop | **-0.50 | **-0.42 | **-0.59 | **-0.51 | 78.9 | 87.8 |
| | AG News | **-0.62 | **-0.54 | **-0.44 | **-0.44 | 50.3 | 59.4 |
| | IMDB | 0.04 | 0.09 | -0.08 | -0.14 | 89.3 | 92.2 |
| | DBpedia | **-0.47 | *-0.27 | **-0.35 | *-0.21 | 27.2 | 33.4 |
| | Emotion | **-0.33 | **-0.42 | **-0.48 | **-0.55 | 29.3 | 31.7 |
| | Tweet Offensive | 0.14 | *0.24 | *-0.20 | -0.03 | 41.6 | 46.2 |
| OPT 30B | Antonyms | **-0.54 | **-0.70 | – | – | – | – |
| | GLUE Cola | -0.05 | 0.03 | -0.13 | 0.02 | 32.2 | 35.5 |
| | Newspop | *-0.23 | *-0.25 | *-0.18 | -0.12 | 60.3 | 66.6 |
| | AG News | **-0.66 | **-0.71 | **-0.81 | **-0.80 | 49.3 | 60.7 |
| | IMDB | -0.06 | *0.17 | 0.04 | **0.22 | 81.6 | 86.1 |
| | DBpedia | **-0.41 | **-0.34 | *-0.21 | *-0.25 | 35.9 | 42.4 |
| | Emotion | 0.00 | -0.03 | 0.18 | 0.13 | 12.3 | 16.2 |
| | Tweet Offensive | **-0.44 | **-0.39 | -0.11 | -0.05 | 54.6 | 60.2 |
| OPT 1.3B | Antonyms | **-0.45 | **-0.53 | – | – | – | – |
| | GLUE Cola | **-0.39 | **-0.36 | -0.09 | *-0.19 | 60.3 | 65.9 |
| | Newspop | **0.33 | *0.21 | -0.07 | -0.07 | 37.6 | 40.3 |
| | AG News | **-0.33 | **-0.29 | **-0.56 | **-0.49 | 31.9 | 37.6 |
| | IMDB | -0.11 | -0.07 | **0.24 | **0.22 | 86.0 | 89.1 |
| | DBpedia | -0.16 | -0.14 | -0.02 | -0.01 | 8.7 | 9.2 |
| | Emotion | 0.08 | 0.08 | **-0.29 | **-0.30 | 7.0 | 9.1 |
| | Tweet Offensive | **-0.42 | **-0.35 | **-0.50 | **-0.38 | 58.6 | 62.6 |

Table 5: Correlation results for the different tasks, with OPT (different sizes) and Bloom. Correlations with $p < 0.05$ are marked with *. Correlations with $p < 0.00625$ (according to Bonferroni correction for multiple hypotheses) are marked with **. Dark and light blue colored cells stand for negative correlations $< -0.2$ and $> -0.2$, respectively. Dark and light orange colored cells stand for positive correlations $> 0.2$ and $< 0.2$, respectively. Average accuracy across all prompts and average accuracy of best 50% prompts are also reported for reference (Avg Acc and Acc 50%, respectively).

scores.

For the word prediction tasks we only report scores, since accuracy in general is less stable, suffers more from the surface form competition (Holtzman et al., 2021), and is usually quite low for these tasks in our setting (the chances the model will generate an exact match of the label are low). Hence, the score of the correct label gives a better estimate of the actual performance of the model.

| Lang | OPT 175B | | Bloom 176B | |
| --- | --- | --- | --- | --- |
| | Pearson | Spear. | Pearson | Spear. |
| ita | -0.44 | -0.57 | -0.37 | -0.63 |
| spa | -0.47 | -0.61 | -0.51 | -0.66 |
| cat | -0.47 | -0.58 | -0.24 | -0.31 |
| fra | -0.48 | -0.57 | -0.48 | -0.64 |
| deu | -0.44 | -0.60 | -0.46 | -0.65 |
| fin | -0.44 | -0.62 | -0.34 | -0.56 |
| por | -0.45 | -0.62 | -0.46 | -0.61 |
| eus | -0.47 | -0.61 | -0.45 | -0.61 |
| tur | -0.44 | -0.62 | -0.33 | -0.62 |
| jpn | – | – | -0.33 | -0.26 |
| arb | – | – | -0.36 | -0.47 |
| rus | – | – | -0.54 | -0.69 |
| kor | – | – | -0.42 | -0.58 |
| ell | – | – | -0.40 | -0.51 |

Table 6: Correlation results for word-level translation, with OPT 175B and Bloom 176B. All correlations are statistically significant also according to Bonferroni correction for multiple hypotheses for OPT ($p < 0.0055$). Same for Bloom ($p < 0.00357$), except for Catalan (Pearson) and Japanese (Spearman).

## 5 Results

**Classification Tasks and Antonym Prediction**
Table 5 depicts the Pearson and Spearman correlation results on the classification tasks and the antonym task, with both OPT 175B and Bloom (two upper blocks). We see that most correlations are negative and statistically significant, as we expect. This validates our hypothesis and shows that in the majority of tasks we indeed get a strong correlation between low perplexity of the prompt and better performance on the task.[10] For each task we also report the average accuracy.

**Word-Level Translation** The results of the word-level translation task are reported in Table 6. Here the correlations are extremely consistent across all languages and across models, with statistical significance for all languages except for Catalan and Japanese (in Bloom).

**Results across Different Model Sizes** We repeat the same experiment with the OPT models of sizes 1.3B and 30B, to investigate whether these correlations are also consistent across model sizes or whether this is a phenomenon we should expect only in large language models. Table 5 (two lower blocks) shows these results for all classification tasks and antonym prediction. We do see that in

---

[10]Repeating the experiments with the length of the prompt instead of perplexity yields weak positive correlations, almost all of which are not statistically significant.

general the trend appears to be the same in the smaller models as well; however, the correlations seem to be slightly weaker. We hypothesize that this might be due to the overall lower performance of these smaller models, making the performance results we use for correlation less stable and reliable. For word-level translation, however, all correlations with the 30B and 1.3B models are similar to those with the 175B model, and are all statistically significant (also after Bonferroni correction for multiple hypotheses).

## 6 Analysis

Next, we further explore the observed relationship between model perplexity and prompt performance. Despite the consistently high correlation between these two factors, the structure of this relationship varies across tasks (Section 6.1). Additionally, we find that the automatically added prompts are high-quality and not a significant source of noise (Section 6.2), and that the best prompts selected by our approach vary across models (Section 6.3).

### 6.1 Visualizing the Relationship between Perplexity and Performance

To visualize the correlations we get between the perplexity and the performance of the prompts across the different settings, we plot a few examples for different tasks and languages. Figures 1 and 2 show some of the results for selected tasks, as detailed in the captions. The negative trend of the correlation is clearly visible in all plots. Interestingly, the structure of the plots for word-level translation are very similar across all the language pairs, suggesting that prompts get consistent perplexity and performance across languages (possibly at different scales). Indeed, the intersection of the 10 lowest perplexity prompts between any two different languages is 8.6 and 8.4 on average (for OPT 175B and Bloom, respectively), which is extremely high. This is not very surprising since we know that the only differences between the prompts in the different languages are the names of the target languages (e.g., The word for "dog" in *French* is "). Additionally, the intersection of 10 prompts with the highest label score between any two different languages is 7 and 6.5 on average (for OPT 175B and Bloom, respectively).

A notable finding that appears in the word-level translation plots is the clear separation between prompts that include or do not include quotation

marks for the label (usually aligns with whether the prompt uses quotation marks for the source word) – three example prompts appear on the plot. Prompts with quotation marks for the words tend to have both lower perplexity and better performance, consistently. We further analyze the results for OPT 175B within clusters (with/without quotations marks). In the cluster with quotation marks, we get negative correlations (in the range of –0.28 to –0.38) that are statistically significant for almost all languages. The correlations within the other cluster are weaker and less significant (this is expected given the overall lower performance of that cluster).

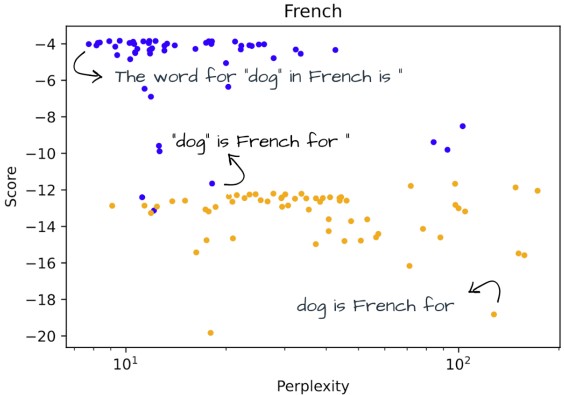

Figure 2: Score of correct label vs. perplexity for the word-level translation task in French with OPT 175B. The $x$ axis is in log scale. The blue points stand for prompts with quotation marks for the words, while the yellow points are of prompts without quotation marks.

## 6.2 Effect of Noisy Prompts

We expect our automatic method for expanding the set of prompts to also introduce some noise. Though our focus is on the lower perplexity prompts, since we want to benefit from this analysis and be able to devise a method for creating better prompts, we do want to make sure that this potential noise is not the cause for the strong correlations we get. In other words, one might claim that some noisy prompts have particularly high perplexity and also perform badly, thus, supporting our hypothesis in an undesirable and uncontrolled manner.

We turn to inspect the 10% highest perplexity prompts in the different tasks and find subjectively that they are not noisy, and are usually valid prompts for the tasks. The 5 highest perplexity prompts for the GLUE Cola task are listed in Table 7 as an example.

| prompt | ppl |
|---|---|
| Is this example correct English usage? | 25.79 |
| Is this example using English correctly? | 25.46 |
| Is this example correct English? | 25.33 |
| Is this the example in correct English? | 25.00 |
| Is English in this example correct? | 24.90 |

Table 7: Example of the 5 highest perplexity prompts for GLUE Cola, using OPT 175B.

| Task | Lang | Before filtering | | After filtering | |
|---|---|---|---|---|---|
| | | Pearson | Spearman | Pearson | Spearman |
| AG News | - | -0.63 | -0.68 | -0.62 | -0.54 |
| WLT | ita | -0.44 | -0.58 | -0.44 | -0.57 |
| | spa | -0.47 | -0.61 | -0.47 | -0.61 |
| | cat | -0.45 | -0.57 | -0.47 | -0.58 |
| | fra | -0.47 | -0.57 | -0.48 | -0.57 |
| | deu | -0.43 | -0.60 | -0.44 | -0.60 |
| | fin | -0.41 | -0.60 | -0.44 | -0.62 |
| | por | -0.43 | -0.61 | -0.45 | -0.62 |
| | eus | -0.45 | -0.60 | -0.47 | -0.61 |
| | tur | -0.43 | -0.61 | -0.44 | -0.62 |

Table 8: Correlations before and after filtering out noisy prompts, with AG News and Word-Level Translation (WLT).

As a sanity check, we choose two tasks: word-level translation and AG News, manually filter out the noisy prompts, and compute the correlations again. The annotation is done by external annotators (NLP researchers) that were presented with the tasks and asked to label whether the prompt is reasonable to use for the task. The new correlations with OPT 175B are reported in Table 8. We find that all correlations remain strong and statistically significant when noise is manually removed from the analysis. We get the same trends with Bloom as well.

## 6.3 Best Performing Prompts

Table 9 lists the 5 lowest perplexity prompts for the task of antonym prediction, as an example. Similar lists for the rest of the tasks are listed in Section B in the Appendix.

A closer look at the lowest perplexity prompts reveals that the intersection of 10 lowest perplex-

| prompt | ppl |
|---|---|
| The following two words are antonyms: "*good*" and " | 10.24 |
| The antonym of the word "*good*" is " | 10.32 |
| The word that has the opposite meaning of the word "*good*" is " | 10.43 |
| The word "*good*" is the antithesis of the word " | 10.85 |
| The word "*good*" is the opposite of the word " | 11.15 |

Table 9: Lowest perplexity prompts for the antonym prediction task, using OPT 175B.

ity prompts between OPT 175B and Bloom is 7.1 on average, across the classification tasks. When looking at the 10 highest accuracy prompts across models we get an average intersection of 3.1 across the classification tasks.

# 7 SPELL: Selecting Prompts by Estimating LM Likelihood

The primary contribution of this work is the analysis of the relationship between prompt perplexity and downstream task performance (Section 5). As one potential application of our findings, we also present a new method, SPELL, for generating and selecting consistently effective prompts.

Assuming a fixed computational budget for finding effective prompts for a given task, and that the search space might be quite large, we devise the following straightforward procedure:

1. Obtain a small set of manually created prompts for the task.

2. Expand the set of prompts with automatic paraphrasing using a LM (e.g., GPT3) and backtranslation (see Section 3).

3. Rank the list of prompts by perplexity (averaged on a representative sample of task inputs, e.g., 1,000).

4. Choose the $k$ (e.g., 3) lowest perplexity prompts.

Using this algorithm, we show empirically that it is best to prioritize experimenting with the lowest perplexity prompts, as they are more stable (exhibit less variation in performance) and perform better than manual prompts on average. This method also does not require any labels for the task, and is applicable to any task, also by non-experts, given example inputs only.

## 7.1 Empirical Validation of SPELL

To show the effectiveness of our method, we report the results we get using SPELL across the different tasks. In Table 10 we report the average accuracy with the manual prompts compared to the average accuracy with the 3 lowest-perplexity prompts, for both OPT 175B and Bloom. Indeed, in most cases, the average accuracy using the 3 lowest perplexity prompts outperforms the average accuracy of the manual prompts, with an average of 1.8 accuracy points across tasks with OPT and 2.3 accuracy

| | OPT | | | Bloom | | |
|---|---|---|---|---|---|---|
| Task | low-ppl | manual | Δ | low-ppl | manual | Δ |
| GLUE Cola | 51.7 | 48.5 | 3.1 | 64.5 | 60.9 | 3.6 |
| Newspop | 80.6 | 70.4 | 10.2 | 90.0 | 80.0 | 10.0 |
| AG News | 68.4 | 61.9 | 6.5 | 51.0 | 63.5 | -12.5 |
| IMDB | 90.4 | 88.9 | 1.4 | 91.3 | 88.8 | 2.5 |
| DBpedia | 46.0 | 51.7 | -5.7 | 31.2 | 30.2 | 1.0 |
| Emotion | 21.6 | 22.6 | -1.1 | 35.8 | 32.1 | 3.6 |
| Tweet Offensive | 48.4 | 50.6 | -2.3 | 48.6 | 40.8 | 7.8 |

Table 10: The average accuracy with the manual prompts (manual) compared to the average accuracy with the 3 lowest-perplexity prompts (low-ppl), for both OPT 175B and Bloom, across tasks.

points with Bloom, demonstrating the effectiveness of our method.

The variability in accuracy of the 3 lowest perplexity prompts is also much lower than that of the manually created prompts: with OPT 175B, the average standard deviation within the 3 lowest perplexity prompts (across tasks) is 5.07, vs. 6.86 for the manual prompts, and with Bloom the gap is much bigger, with an average of 2.6 for the 3 lowest perplexity prompts vs. 7.47 for the manual ones.[11] This further shows that SPELL is more stable and reliable compared to using an arbitrary set of manually created prompts. SPELL sets the stage for further development in this direction, and serves as an initial indication of the benefits of involving perplexity estimation in the process of generating effective prompts.

# 8 Related Work

**Relation between performance and training data** Previous work looking directly into the relation between the training data and the performance is limited. Razeghi et al. (2022) study numeric deduction tasks, and examine the correlations between the model performance on specific test instances and the frequency of terms from those instances in the pretraining data. They find that the models are more accurate on instances whose terms are more prevalent in the training data. Additionally, Han and Tsvetkov (2022) propose a method to effectively identify a very small subset of pretraining data that directly supports the model in performing a specific task. Elazar et al. (2022) use causal inference to measure the effect of pretraining data statistics on factual knowledge performance, and

---
[11]We also calculate the standard deviation when using the same amount of low-perplexity prompts as in the manual prompts set for each task and get averages of 6.32 and 3.78 for OPT 175B and Bloom, respectively.

Kandpal et al. (2022) show correlational and causal relationships between accuracy and relevant document count (from training data) for QA datasets.

**Prompt tuning and analysis**   There is a very rich line of work trying to find prompts automatically. Shin et al. (2020) present an automated method to create discrete prompts for a diverse set of tasks, based on a gradient-guided search, and they demonstrate their method on masked LMs. Other work also focuses on discrete prompts, aiming to improve zero-shot performance (Gao et al., 2021; Le Scao and Rush, 2021; Deng et al., 2022; Shi et al., 2022), or trains continuous prompts (Li and Liang, 2021; Lester et al., 2021; Qin and Eisner, 2021).

On top of works that suggest a variety of methods for creating better prompts, some work also analyzes those prompts to try and get some insights about them: Khashabi et al. (2022a) find that model performance is highly sensitive to small changes in wordings and Khashabi et al. (2022b) point to a surprising disconnect between continuous and discrete prompts.

## 9   Conclusion

We investigate the phenomenon where some prompts perform better than others despite appearing similar to the human users of LMs. Specifically, we hypothesize that the perplexity of a prompt under a given LM is closely tied to its task performance. We test this theory on a large number of tasks and autoregressive LMs, and the resulting correlation study validates our hypothesis. Further analysis of this relationship demonstrates that the best prompts differ across models, highlighting the importance of model-specific analysis, and that the underlying structure of the relationship between perplexity and performance varies across tasks.

In light of these findings, we then propose a method, SPELL, to help users find well-performing prompts for new tasks. Empirical validation of the proposed procedure shows that SPELL generates effective prompts with low variability in performance, and produces small gains of 1.8 (2.3) accuracy points with OPT (Bloom) over manual prompts. We therefore conclude that SPELL provides a general and interpretable approach for applying LMs to new tasks while requiring minimal human effort, and no labels.

## Limitations

**Searching for human-readable prompts**   We limit our search space to human-readable prompts that are fluent and accurately describe the task at hand, as we are primarily motivated in understanding why some relevant prompts work better than others. We do this by using manually created prompts and their automatically created paraphrases. Our findings may not hold when the possible prompt space is expanded to include any token sequence; we leave this direction to future work.

**Generality of our analysis and of the SPELL method**   We perform our analysis on and build our method around specific models, namely OPT and Bloom. Additionally, our study is limited to the specific tasks we experiment with and to English. It is possible that our analysis and SPELL method do not generalize to other pretrained models or tasks; however, we consider models of various sizes and from different sources, and a wide range of tasks to mitigate this risk.

## Acknowledgements

We thank Alisa Liu and Orevaoghene Ahia for their help in annotating noisy prompts. We also thank the reviewers for their valuable comments on the paper.

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

## A  Manually Created Prompts

Table 11 lists the manually created prompts we use for the different tasks. We manually add, remove and edit prompts for some of these tasks, to make them fit for our setting. For example, the following prompt for AG News, taken from Promptsource, does not fit our setting: *Would you recommend the following article to a politician, an athlete, business executive, or a scientist?*

## B  Lowest Perplexity Prompts

Table 12 lists the 5 lowest perplexity prompts for each task, using OPT 175B.

| Task | Manual Prompts |
|------|----------------|
| Antonyms | The antonym of the word "good" is "
The opposite meaning of the word "good" is "
"Good" is the opposite of "
"Good" is the negation of "
The following are opposites of each other: "good" and "
The word "good" contradicts the word "
The antonym of the word good is
The opposite meaning of the word good is
Good is the opposite of
Good is the negation of
The following are opposites of each other: good and
The word good contradicts the word |
| GLUE Cola | Does the this sentence make sense and use correct English?
Is this example grammatically correct and sensible?
Does this sentence make sense and is it grammatically correct?
Does this example use correct English? |
| Newspop | What is the article about?
What is this news about?
What is the topic of this news piece?
What does this article discuss?
What is the topic of this sentence?
What category does the article belong to?
Pick one category for this news piece.
Pick the category that fits the text.
The article refers to which category?
What topic does the article belong to?
What category fits this article?
What topic does this news piece belong to?
Choose the correct category for this article. |
| AG News | What label best describes this news article?
What is this piece of news regarding?
Which newspaper section would this article likely appear in?
What topic is this news article about? |
| IMDB | This movie review expresses what sentiment?
Did the reviewer find this movie good or bad?
Is this review positive or negative?
How does the viewer feel about the movie?
What sentiment does the writer express for the movie?
What sentiment is expressed for the movie?
What is the sentiment expressed in this text?
Did the reviewer enjoy the movie?
What is the sentiment expressed by the reviewer for the movie?
How does the reviewer feel about the movie? |
| DBpedia | What category does the paragraph belong to?
Pick one category for the text.
Pick the category that fits the text.
The text refers to which category?
What category does the title belong to?
What category fits this text?
What topic does this text belong to?
Choose the correct category for the text. |
| Emotion | What is the emotion expressed in this message?
What emotion does this message express?
How will you feel about the message?
What emotion does the writer express for the message? |
| Tweet Offensive | Is this tweet offensive?
Can the tweet be removed for being offensive?
Is the author's tweet offensive?
Task: Identify if the tweet or text is offensive.
Is this an offensive tweet? |
| Word-Level Translation | The translation of the word "dog" to French is "
The translation of the word dog to French is
The word "dog" in French is "
"dog" (In French: "
Translate the word dog into French:
The translation of dog to French is
"dog" (French: "
The word dog in French is
Translate the word "dog" into French: "
dog (In French:
dog (French:
The translation of "dog" to French is " |

Table 11: The set of manually created prompts for each task.

| Task | Lowest Perplexity Prompts | Perplexity |
|---|---|---|
| Antonyms | The following two words are antonyms: "good" and " | 10.24 |
| | The antonym of the word "good" is " | 10.32 |
| | The word that has the opposite meaning of the word "good" is " | 10.43 |
| | The word "good" is the antithesis of the word " | 10.85 |
| | The word "good" is the opposite of the word " | 11.15 |
| GLUE Cola | Is this an example of the proper use of the English language? | 11.63 |
| | Does the sentence make sense and does it follow the rules of grammar? | 11.76 |
| | Is this sentence an example of the correct use of the English language? | 12.10 |
| | Does this sentence make sense and is it grammatically correct? | 12.15 |
| | Is this sentence grammatically correct and does it make sense? | 12.68 |
| Newspop | What is the main subject of the article? | 10.01 |
| | What is the main topic of the article? | 10.01 |
| | What is the subject matter of the article? | 10.17 |
| | What is the subject of the article? | 10.21 |
| | What is the main idea of this article? | 10.21 |
| AG News | In what section of the newspaper would you expect to find this article? | 7.51 |
| | In which section of the newspaper would you expect to find this article? | 7.52 |
| | In which section of the newspaper would this article be most likely to appear? | 7.60 |
| | In what section of the newspaper do you expect to find this article? | 7.80 |
| | In what section of the newspaper would this article most likely appear? | 7.87 |
| IMDB | What is the opinion of the review? Is it positive or negative? | 7.19 |
| | Is this a positive or negative review? | 7.31 |
| | What do you think of the movie? | 7.33 |
| | What do you think of the film? | 7.35 |
| | Is that a positive or a negative? | 7.35 |
| DBpedia | What is the category to which the text refers? | 8.99 |
| | What is the subject of the text? | 9.15 |
| | What category does the title belong to? | 9.18 |
| | Which category does the text refer to? | 9.19 |
| | What is the subject of this text? | 9.20 |
| Emotion | How do you feel when you hear this message? | 12.72 |
| | What is the writer's emotional reaction to this news? | 13.18 |
| | What is the emotion expressed in this message? | 13.20 |
| | How does this message make you feel? | 13.32 |
| | How do you feel about this message? | 13.50 |
| Tweet Offensive | If someone said this to you, would you be offended? | 13.00 |
| | If someone said that to you, would you be offended? | 13.10 |
| | Would you be offended if someone said that to you? | 13.73 |
| | Would it offend you if someone said that to you? | 14.79 |
| | If someone told you that, would you be offended? | 14.93 |
| Word-Level Translation | The word for "dog" in French is " | 7.73 |
| | The French word for "dog" is " | 8.16 |
| | The French translation of the word "dog" is " | 8.24 |
| | The translation of the word "dog" in French is " | 8.35 |
| | The translation of the word "dog" into French is " | 8.91 |

Table 12: The 5 lowest perplexity prompts for each task, using OPT 175B.