# OpenReview forum: "Demystifying Prompts in Language Models via Perplexity Estimation"
_EMNLP/2023/Conference — EMNLP 2023 Findings_

### Official Review · Reviewer_nmvT · 2023-08-03

**Soundness:** 4

**Excitement:**

4: Strong: This paper deepens the understanding of some phenomenon or lowers the barriers to an existing research direction.

**Paper Topic And Main Contributions:**

This paper puts the following hypothesis to the test: the performance of a prompt is predicted by the extent to which the model is familiar with the language it contains. The authors use perplexity as the proxy measure of familiarity of the LM with the prompt language. Based on the empirical findings, lower perplexity does correlate with better performance. Furthermore, the authors use this insight to propose a prompt selection method based on the perplexity measure.

**Questions For The Authors:**

- While I assumed that correlations with score could be more significant compared to correlations with accuracy (but in the same direction), this seems not to be the case in Table 3. Can you provide more explanation and comparison between the two?
- The overall accuracy of the models on the tasks chosen is quite low. I wonder how much of these results extend to models fine-tuned with instructions, which seems to be the de-facto standard these days, do the authors have any opinion on this?

**Reasons To Accept:**

- Understanding why some prompts might work better than others and finding systematic ways to find better prompts is in fact a very important research direction, especially since prompting (in-context learning) is becoming the de-facto standard for adapting LMs to downstream tasks. This paper addresses exactly this research question.
- The experiments and analysis are carefully designed while controlling for confounders.
- The paper is generally well-written and easily understandable.

**Reasons To Reject:**

- From line 126 onwards, the authors state that lower perplexity correlated with better performance. A possible confounder here could be the relevance of the prompts. While in section 6.2 the authors investigate the effect of noisy prompts, I wonder if the relevance of the prompts to the task can be factored in systematically earlier in experiments in section 5 (like Table 3 results).
- Lines 480-482 compare the variability in accuracy between the two settings, however, the number of prompts assessed in these two settings are not equal, doesn't it introduce some bias?
- In Table 8, for the Emotion task, we see that the delta is negative, this is true while in Table 3 and for the same OPT model and Emotion task, the correlation was negative. I wonder how this could be the case.

**Reproducibility:**

4: Could mostly reproduce the results, but there may be some variation because of sample variance or minor variations in their interpretation of the protocol or method.

**Reviewer Confidence:**

4: Quite sure. I tried to check the important points carefully. It's unlikely, though conceivable, that I missed something that should affect my ratings.

---

> ### Author Rebuttal · Authors · 2023-08-28
>
> Thanks a lot for your positive feedback!
>
> 1. Relevance of the prompt:
> Note that the space of prompts we are dealing with has only prompts that are presumably related to the task (hand written), and those that are paraphrased out of them. So except for noise, with which we also deal later - all prompts are relevant to the task. A more fine grained definition of relevance could be explored here, but these distinctions are subjective and difficult to quantify. We do believe that other factors than perplexity can be predictive of the variability among prompts.
>
> 2. Variability in accuracy:
> Thanks for raising this point. We did see consistently good performance also among 5 and 10 lowest-ppl prompts (not only 3). To make sure that the samples are of the same size, we will choose the same amount of prompts for each task and add those results to the final version as well.
>
> 3. Emotion task:
> We believe that when the delta is negative, it is due to hand written prompts that happened to work well. This is directly related to one of the points we make in the paper: with hand-written prompts we are not always guaranteed to get good performance, and whether this is the case for a particular prompt is very difficult to predict from the text of the prompt.
>
> QA:
> Correlation with accuracy vs score: Indeed, there is no clear “winner” between the two, though both show consistent correlations with the performance. A possible explanation for the difference could be the variability of the scores of the different labels, for example: if the scores vary a lot, you could get a weaker correlation even if the rankings are relatively good (high accuracy). This may also be related to the inherent difference between the two metrics, accuracy being brittle and discrete vs. LM scores that are continuous. We see this as a very interesting point for future work.
>
> QB:
> Instruction-tuned models: We agree that extending our experiments to instruction-tuned models would be very interesting! We expect to see similar trends there as well, but it’s an empirical question.

---

### Official Review · Reviewer_feBz · 2023-08-04

**Paper Topic And Main Contributions:** 1. This paper investigates the resear…
**Soundness:** 4

**Excitement:**

3: Ambivalent: It has merits (e.g., it reports state-of-the-art results, the idea is nice), but there are key weaknesses (e.g., it describes incremental work), and it can significantly benefit from another round of revision. However, I won't object to accepting it if my co-reviewers champion it.

**Questions For The Authors:**

* Question A: How do you prove it is perplexity but not other factors that influence the performance? In Table 1, it seems that the best prompt describes the task target the most precisely. What if we construct some prompts with higher perplexities but containing more accurate task descriptions? Let us say: Among all prompts that obtain high perplexity, we manually choose the ones that give accurate task descriptions. How do they perform on downstream tasks then?

* Question B: For classification tasks, did you explicitly provide label words? How do you choose label words then?

**Reasons To Accept:**

* This work presents an analysis of the correlation between LM's perplexity with prompts and downstream performances. The analysis has some merits for future research.

* The proposed SPELL shows performance gains on some datasets, especially for Newspop (+10 accuracy points across two LMs).

**Reasons To Reject:**

* The problem setting is not well-established. As this paper focuses on perplexity, there are other factors other than perplexity that could influence downstream performances. These factors should also be discussed in the paper; otherwise, the authors cannot ensure that it is perplexity really matters. More details are in Question A in the next section.

* This paper conduct experiments on word translation and text classification. The two tasks somehow have limited coverage. It would be better if more tasks could be adopted, especially some more complex tasks like planning and reasoning. With more complex tasks, the differences among prompts would be more significant, providing more generalizable experimental conclusions.

* The performance gains of SPELL are not consistent across datasets and LMs, raising concern about whether the proposed method could work in a broader scope.

**Reproducibility:**

4: Could mostly reproduce the results, but there may be some variation because of sample variance or minor variations in their interpretation of the protocol or method.

**Reviewer Confidence:**

3: Pretty sure, but there's a chance I missed something. Although I have a good feel for this area in general, I did not carefully check the paper's details, e.g., the math, experimental design, or novelty.

**Typos Grammar Style And Presentation Improvements:**

Presentation Improvements:
  * Table 1 and Figure 2, would be better if on top of the page.

---

> ### Author Rebuttal · Authors · 2023-08-28
>
> Thanks a lot for your positive feedback!
>
> 1. Other possible factors (QA):
> Our main finding is that perplexity is a predictive signal for the quality of a prompt (rather than a causal factor). We definitely agree that there are other potential factors that likely affect the success of prompts, and that perplexity does not explain all of the variability we see among different prompts. Here we focus specifically on perplexity, and as we show experimentally across 9 tasks and 4 models, it is an important predictive factor and is strongly correlated with the performance of the prompt.
> As for testing the effect of prompt relevance: we only consider prompts that are presumably related to the task (hand written), and those that are paraphrased out of them. The more fine-grained definition of relevance suggested may affect performance, but these distinctions are subjective and difficult to quantify.
>
> 2. Variety of tasks:
> In this paper we try, for the first time, to explore the correlation between the perplexity of the prompt and its performance. When designing the experiments, we wanted to keep the setting basic, in order to not introduce confounding factors into our analysis. Using a zero-shot setting, with somewhat shorter prompts and a well defined set of tasks, helped us achieve this goal. Nonetheless, we do experiment with 4 different models across 9 different tasks.
>
> 3. SPELL:
> As stated in the paper, we want to highlight again that while we believe that the SPELL method is a simple and effective one for getting improved performance by generating and choosing better prompts, this is not the main contribution of our work. Our key findings are the analysis of the differences between different prompts, and our new characterization of what makes a good prompt. This analysis allows us to establish correlations of the effectiveness of a prompt with its perplexity, which in turn allows us to introduce a simple new method to find better prompts. Additionally, an important outcome of SPELL is not only improved performance but also prompts that are less variable in their performance.
>
> QB:
> we do provide label words in the prompt - we simply list the possible answers as defined by the dataset itself (e.g. Sports, Science and technology, World politics, Business for the dataset of ag_news). We will make this clearer in the paper.
>
> We will make changes to Tab 1 and Fig 2 as suggested, thank you!

---

### Official Review · Reviewer_ZmV4 · 2023-08-05

**Soundness:** 3

**Excitement:**

4: Strong: This paper deepens the understanding of some phenomenon or lowers the barriers to an existing research direction.

**Paper Topic And Main Contributions:**

The sensitivity of prompt-based learning to varying templates often results in a wide range of performance levels. This study seeks to understand why certain prompts outperform others by examining the concept of "familiarity". The authors use perplexity estimation to express this concept, ultimately finding that prompts with lower perplexity tend to yield better results. To build upon this discovery, they introduce a methodology for automatically adjusting the initial prompt template, and subsequently choose the most efficient template using perplexity estimation. Through empirical testing and evaluation, it is demonstrated that this approach of perplexity estimation is straightforward yet potent for enhancing zero-shot learning performance.

**Reasons To Accept:**

1. The proposed perplexity estimation approach demonstrates wide applicability in reducing the variance of zero-shot learning performance, particularly in multilingual tasks, as evidenced by the strong correlation between perplexity estimation scores and task performance.
2. The authors introduced a framework that enables automatic rephrasing and adjustment of the seed prompt to enhance compatibility with language models, resulting in improved performance.
3. Through the authors' perplexity estimation method, it is revealed that certain prompts, despite having similar semantics and appearing acceptable to humans, can lead to poor performance. These prompts can be identified using the perplexity estimation approach, providing valuable insights for prompt learning debugging without reliance on external knowledge.

**Reasons To Reject:**

1. While the authors assert that perplexity estimation has the potential for broad application across various tasks, its significance is primarily evident in multilingual tasks. However, the correlation in classification tasks provides less support for this hypothesis.

2. The authors concentrate on zero-shot settings across all tasks, which can pose challenges for smaller language models (such as 1.3B and 30B models) in specific tasks like CoLA and Tweet Offensive. Notably, in these challenging tasks, the performance and correlation may not exhibit a consistent pattern, indicating potential limitations of perplexity estimation.

**Reproducibility:**

5: Could easily reproduce the results.

**Reviewer Confidence:**

3: Pretty sure, but there's a chance I missed something. Although I have a good feel for this area in general, I did not carefully check the paper's details, e.g., the math, experimental design, or novelty.

---

> ### Author Rebuttal · Authors · 2023-08-28
>
> Thanks a lot for your positive feedback!
>
> 1. Correlations of classification tasks vs. word-level translation:
> Indeed, the correlations we get for the word-level translation task tend to be higher. Note however that this is true also for the antonyms task, which is not multilingual. We do not attribute the gap to the multilinguality of the translation task, but to the nature of it - both the word level translation and the antonyms tasks are somewhat easier for a language model (in that they are word predictions and therefore closer to the objective of an LM), and we believe that this is what’s causing the correlations to be stronger for this type of task.
>
> 2. Performance with smaller models:
> We agree that smaller models struggle with the tasks we experiment with, and it’s a known fact that larger models do much better in general, and specifically in zero-shot settings. We expect the 175B models to be more reliable for our set of experiments, and we added the smaller models mainly as a sanity check for lower scale as well.

---

### Meta-Review · Area_Chair_sgzw · 2023-09-20

**Recommendation:** 4

**Metareview:**

This work proposes and investigates the hypothesis that the likelihood assigned by a language model to a task-specifying prompt is a useful predictor of the performance of that model on the task that the prompt specifies, such that among intuitively similar prompts, one can/should pick the highest-likelihood prompt under the model to maximize performance in expectation. The reviewers were overall positive, despite some concerns about the scope of experiments (no instruction tuned / RLHF’d models) as well as uncertainty about how far the hypothesis can hold true (as perplexity alone certainly isn’t the only factor.) Overall, it seems that the claims of the authors are interesting enough and backed up enough by their experiments for a good empirical takeaway.

The reviewers collectively had concerns as to the seemingly causal claims, or at least rather strong correlational claims, that lower prompt perplexity implies better few-shot performance, whereas certainly this is only true within a certain (even if just intuitive) bound of task-relevance or task-descriptive quality; one can likely always find prompts with increasingly high probability under the model with increasingly little to do of the task of interest, which would not work. This is an important nuance that I encourage the authors to engage more seriously with, starting in the abstract. Another related nuance that deserves mention is that this work deals exclusively with the perplexity of the prompt under the model of interest, which is related to the particularities of how the model has learned (or failed to learn) and is distinct, e.g., from the “true” probability/perplexity of the prompt under the training distribution (which is mentioned in the motivation).

Thanks for doing Bonferroni correction, authors!

---

### Decision · Program_Chairs · 2023-10-07

**Decision:**

Accept-Findings

**Comment:**

This work proposes and investigates the hypothesis that the likelihood assigned by a language model to a task-specifying prompt is a useful predictor of the performance of that model on the task that the prompt specifies, such that among intuitively similar prompts, one can/should pick the highest-likelihood prompt under the model to maximize performance in expectation. The reviewers were overall positive, despite some concerns about the scope of experiments (no instruction tuned / RLHF’d models) as well as uncertainty about how far the hypothesis can hold true (as perplexity alone certainly isn’t the only factor.) Overall, it seems that the claims of the authors are interesting enough and backed up enough by their experiments for a good empirical takeaway.

The reviewers collectively had concerns as to the seemingly causal claims, or at least rather strong correlational claims, that lower prompt perplexity implies better few-shot performance, whereas certainly this is only true within a certain (even if just intuitive) bound of task-relevance or task-descriptive quality; one can likely always find prompts with increasingly high probability under the model with increasingly little to do of the task of interest, which would not work. This is an important nuance that I encourage the authors to engage more seriously with, starting in the abstract. Another related nuance that deserves mention is that this work deals exclusively with the perplexity of the prompt under the model of interest, which is related to the particularities of how the model has learned (or failed to learn) and is distinct, e.g., from the “true” probability/perplexity of the prompt under the training distribution (which is mentioned in the motivation).

Thanks for doing Bonferroni correction, authors!